# Optimized Classification of Intelligent Reflecting Surface (IRS)-Enabled GEO Satellite Signals

**DOI:** 10.3390/s23084173

**Published:** 2023-04-21

**Authors:** Mamoona Jamil, Mubashar Sarfraz, Sajjad A. Ghauri, Muhammad Asghar Khan, Mohamed Marey, Khaled Mohamad Almustafa, Hala Mostafa

**Affiliations:** 1School of Engineering & Applied Sciences, ISRA University, Islamabad 46000, Pakistan; 2Department of Electrical Engineering, NUML, Islamabad 44000, Pakistan; 3Hamdard Institute of Engineering and Technology, Hamdard University, Islamabad 44000, Pakistan; 4Smart Systems Engineering Laboratory, College of Engineering, Prince Sultan University, Rafha Street, P.O. Box 66833, Riyadh 11586, Saudi Arabia; 5Department of Information Systems, College of Computer and Information Sciences, Prince Sultan University, Riyadh 11586, Saudi Arabia; 6Department of Information Technology, College of Computer and Information Sciences, Princess Nourah bint Abdulrahman University, P.O. Box 84428, Riyadh 11671, Saudi Arabia

**Keywords:** modulation classification, Gabor filter network, intelligent reflecting surface, GEO satellite signals, global and local search methods

## Abstract

The intelligent reflecting surface (IRS) is a cutting-edge technology for cost-effectively achieving future spectrum- and energy-efficient wireless communication. In particular, an IRS comprises many low-cost passive devices that can independently reflect the incident signal with a configurable phase shift to produce three-dimensional (3D) passive beamforming without transmitting Radio-Frequency (RF) chains. Thus, the IRS can be utilized to greatly improve wireless channel conditions and increase the dependability of communication systems. This article proposes a scheme for an IRS-equipped GEO satellite signal with proper channel modeling and system characterization. Gabor filter networks (GFNs) are jointly proposed for the extraction of distinct features and the classification of these features. Hybrid optimal functions are used to solve the estimated classification problem, and a simulation setup was designed along with proper channel modeling. The experimental results show that the proposed IRS-based methodology provides higher classification accuracy than the benchmark without the IRS methodology.

## 1. Introduction

Next-generation communication technologies offer massive connectivity, high data rates, low latency, and higher levels of service quality. Thanks to recent developments in contemporary aerospace technology, satellite communications can now provide services in various application scenarios, including broadcasting, navigation, and disaster aid. However, the wireless environment is conventionally seen as a variable entity whose uncontrolled reflections, refractions, and interference degrade signal quality. Undeniably, the achievable data rate and performance reliability have reached a saturation point, despite the development of many physical layer techniques, such as advanced modulation/demodulation and precoding/decoding schemes, typically at the endpoints of communication links to compensate for these negative effects. Substantial performance increases are anticipated when the wireless environment is included as an extra component for optimization. This is made feasible by using the innovative concept of reconfigurable intelligent surfaces (RISs) or intelligent reflecting surfaces (IRSs) [1], which are capable of reconfiguring the wireless propagation environment into a transmission medium with more desired features. In addition, IRS-enabled GEO satellite signals can enhance worldwide coverage and expand network capacity, thereby providing crucial support for rapidly expanding Internet of Things (IoT) applications [2]. The integration of IRSs with satellite communication technology also accomplishes the objectives of both the existing Fifth-Generation (5G) and the forthcoming Sixth-Generation (6G) communication networks [3]. Being a low-cost planar surface design with several reflecting components, IRSs can intelligently reconfigure the incident signal propagation by coordinating the reflection coefficient of each element [4].

IRS-assisted communication systems are undergoing preliminary research to improve their spectrum and energy efficiency. However, satellite downlink communications are prone to eavesdropping due to the openness of wireless communications and the vast beam area. In addition, interference difficulties grow more severe since the satellite communication system and terrestrial network use the same frequency range. Recent IRS advancements provide a good chance to address security and interference concerns. The IRS can prevent eavesdropping on the satellite downlink broadcast by reflecting the common-spectrum friendly interference from the terrestrial network. The IRS comprises many passive reflecting elements that could cleverly reroute a signal’s path to the receiver. The incident-radiation-shaping IRS concept states that the phase of the incident signal from the transmitter may be intelligently moved toward the receiver without requiring additional power.

Modulation classification can be crucial in communication for civilian and military applications to detect the received information [5,6,7,8,9].

Next-generation satellite networks are expected to adapt their modulation formats dynamically [10] in response to link conditions and terminal equipment configurations to meet the requirements of various terminal systems and applications. The development of hybrid transmission systems created a challenge for classifying the signals [11,12].

Furthermore, the channel conditions are vital in precisely detecting the transmitted information [13]. Therefore, the classification accuracy can be improved by improving the channel quality. IRSs have recently been proven to improve the channel quality and maximize the coverage region by precisely adjusting the phase shifts of their constituent elements to meet a particular demand [14,15]. The signals that IRSs reflect can be integrated with those that transit through other channels, constructively or destructively, to increase the intended signal strength at the receiver or decrease interference. However, the IRS operates in full duplex compared to the other relays, with no additional power for interference cancellation or amplification [16].

Gabor filters have a wide range of applications, particularly image and texture analysis, because of their exceptional feature extraction capabilities [17,18,19,20]. Consequently, the Gabor filtering network (GFN) [21] technique was presented to extract unique attributes required to distinguish modulation forms. Furthermore, a genetic algorithm (GA) [22] was applied to optimize the weights of the GFN to improve classification accuracy. Based on these discussions, this research provides a strategy for integrating the GEO satellite signal with an IRS, complete with the appropriate channel modeling and system characterization. Gabor filter networks (GFNs) are proposed for extracting and categorizing different features. Hybrid optimal functions are employed to solve the approximated classification issue, and a simulation setup and appropriate channel modeling have been developed. Key contributions to the research are mentioned in the section that follows.

### 1.1. Contributions of the Article

In this paper, we propose a classification scheme that, to our knowledge, has never been modeled for classifying IRS-enabled GEO satellite signals, namely, M-QAM and M-PSK signals. The classification scenario is based on a Gabor filter network (GFN). The GFN has three parameters, i.e., shift, scale, and modulation parameters; these parameters are tuned and optimized for the accurate classification of the modulation formats. First, we describe the system model that characterizes the signal received at the receiver end. Then, GFN parameters are optimized using a genetic algorithm and its local search variants, such as GA-Fminsearch, GA-Pattern Search, GA-Fminunc, and GA-Fmincon. The genetic algorithm-assisted GFN provides the optimized values of GFN parameters for classifying M-QAM and M-PSK signals. The simulations investigate the performance trade-off between various approaches in the system. The impact of increasing the sample size is also explored. The major contributions include the following:We introduce a scheme for IRS-equipped GEO satellite signals with proper channel modeling and system characterization.The parameters of the GFN are analyzed and optimized for the classification of M-QAM and M-PSK signals.Finally, a performance analysis of the proposed classifier structure was conducted with and without the IRS.The findings show that the classification accuracy of the proposed IRS-based methodology is better than that of the benchmark without the IRS-based methodology.

### 1.2. Organization of Article

The remainder of this article is arranged as follows. Section 2 presents a detailed literature review of the existing techniques. The proposed IRS-enabled architecture is derived in Section 3, and the problem is formulated for the GFN-based modulation classification. The performance evaluation of the proposed problem is reported and discussed in Section 4, where the graphical and numerical analyses are presented. Finally, the proposed problem is concluded in the last section, i.e., Section 5.

## 2. Related Work

The academic research prioritizes modulation classification, focusing on robustness, computational efficiency, and better channel recognition for all binary schemes. Massive work in the future will be on new generations of wireless communication to serve three main areas. The first is to enhance cellular broadband, with enormous work on machine-type communication. The last thing is reliability in the sense of its working with new technology in communication, such as the rate in NOMA [23], though this feature is mainly used in applications on the internet relating to mobile communication systems such as Unmanned Aerial Vehicles (UAVs) [3,24]. In satellite communication, water efficiently works to secure user data and broadcast base power consumption and acts as an agent of communication, in which the bulk of data are received at the downlink receiver side [25].

The study of the literature on 5G reveals a lot of barriers created in the 5th generation; for example, the range of cellular networks and data rate are uneven, and the battery life always involves trade-offs. Besides these, users face the challenge of utilizing more power to access signal connections [26] directly. The cost-effect minimization of IRS elements, such as broadcasting devices/sensors, is possible by replacing the IRS with IRS-AP. The cost of components is reduced by using a pattern-based receiving channel at the receiver side, where the data sent by the user are reflected with the help of the IRS. In this way, prior information topology was used to estimate the cost of each affected element for ON/OFF pushing signals in the pattern reflector approach [27,28].

Because of restricted resources and privacy concerns, the authors in [29] developed a mobile edge cloud and a federated learning framework to classify satellite signals. With the help of higher-order cumulants, the features of the satellite signals were extracted, and neural networks were used to perform the classification [11]. The performance of various machine learning algorithms for classifying satellite signals was compared with that of a CNN, and the proposed CNN’s dominance is presented in [12]. The authors in [30] proposed a hybrid feature extraction network that combines spatial and temporal features for classification. The two feature extraction networks are intended to map wireless communication signals to temporal and spatial subspaces. A hybrid loss function was presented to train the proposed network more effectively, promoting more inter-class signal separability in the two feature spaces. In few-shot AMC tasks, experimental findings proved the usefulness and resilience of the proposed network.

For an unknown frequency of selective channels, the authors in [31] proposed a three-dimensional convolutional neural network to classify multiple-input and multiple-output orthogonal frequency division multiplexing signals. Complex envelope samples of a burst signal collected by many antennas are decomposed at the receiver into in-phase and quadrature samples before being organized into a high-dimensional data array. MONet achieved a classification accuracy of over 95 percent at 0 dB SNR under various channel impairments and demonstrated resilience with various MIMO antenna topologies using simulations.

An eye diagram is a signal representation that displays important factors, such as timing jitter and inter-symbol interference. The eye diagram comprises essential elements that might be utilized for spectrum awareness activities. Deep learning and an eye diagram were used in [32] for classification. The authors presented a new approach by integrating the meta-heuristic technique’s synergy with Gabor feature extraction, commonly used in texture analysis. Gabor filters to extract features, which are then improved using the cuckoo search and genetic algorithm to improve the classification procedure’s efficiency [33,34].

In [35], the authors introduce a novel approach to generating super-cumulants by fusing the traditional mathematical method of linearly computing combinations of cumulants with a genetic algorithm (GA). The super-cumulants allow the K-nearest neighbor technique to classify five digital modulation schemes when applied to fading channels.

The authors deployed a multi-task learning framework based on deep neural networks to perform modulation and signal classification tasks concurrently while considering diverse wireless signals in the electromagnetic spectrum, such as radar and communication waveforms. The proposed design uses the tight link between the two objectives to increase the classification accuracy and learning efficiency by employing a lightweight neural network model [36]. In addition to experimental evaluations of the model using samples from the air, the authors provided first-hand knowledge of model compression and a deep learning pipeline for deployment on resource-constrained edge devices. Using the proposed paradigm, they demonstrated considerable computational, memory, and accuracy improvements over two reference designs. In addition to simulating a lightweight MTL model appropriate for resource-constrained embedded radio systems, the authors modeled a lightweight one.

The authors in [37] presented the classification of modulation formats using random forest. The classification algorithm’s performance depends on the extracted features, even though some of the collected characteristics are noisy, irrelevant, and redundant. Inappropriate or noisy features will have a negative impact on the profile performance. Increasing the number of retrieved features would directly or greatly increase the training time of various pattern recognition algorithms. Various deep-learning-based classifications are proposed in [38,39,40,41]. A detailed review of the classification of modulation formats using deep learning with their datasets and challenges is presented in [42]. In addition, the authors conducted extensive tests to assess the state-of-the-art models for single-input, single-output (SISO) systems in terms of accuracy and complexity. They suggested using DL-AMR in a novel multiple-input multiple-output (MIMO) scenario with precoding. Finally, existing obstacles and potential future avenues of study are highlighted.

## 3. Proposed Classifier

This section details the system model, the Gabor filter network, and the optimization of Gabor parameters and weights. First, the mathematical model of the proposed signal flow from the satellite to the ground station with and without the IRS is developed. This type of channel modeling for modulation classification purposes has yet to be reported in the literature. After channel modeling, the received signal is fed to the classifier structure; this research uses the Gabor filter network to extract distinct features. After successfully extracting the Gabor features, these features are optimized using the genetic algorithm (GA) and its hybrid local search variants, i.e., GA-Fminsearch, GA-Pattern Search, GA-Fminunc, and GA-Fmincon. This section discusses the communication model, the GFN, and the optimization of GFN parameters in detail.

### 3.1. System Model

As depicted in Figure 1, we assume a satellite communication network in which a GEO satellite terminal *S* aims to send a signal to a ground mobile terminal, which, in our case, acts as a receiver *R*. Line-of-Sight (LOS) communication from satellite terminals is not always accessible; as a result, an IRS unit is installed on the building’s wall to help transmit the satellite signal to the receiver. The IRS has *K* reflecting components, and the receiver receives numerous beams. We assume the receiver receives the signal from the satellite terminal using a single antenna. The receiver end considers the channel state information (CSI) entirely known.

The channels of different links in the proposed system model are *S*$\overset{}{\to }$IRS and IRS$\overset{}{\to }$*R*. The channel gain from *S*$\overset{}{\to }$IRS is denoted by $\zeta_{q}\in Z^{Q\times 1}$, with $\zeta_{q}$ = $\left[\zeta_{1},\zeta_{2},...,\zeta_{q},...,\zeta_{Q}\right]$, where $\zeta_{q}$ is the channel gain from *S* to IRS. The channel gain $\zeta_{q}$ can be defined as [43]:(1)$$\zeta_{q}=e^{-i\psi }\frac{\sqrt{G^{S}}}{4\pi \frac{d_{\zeta }}{\lambda }}$$
where $G^{S}$ is the gain of *S*, $d_{\zeta }$ is the distance from *S* to IRS, and ψ is the phase of *S*. The channel from IRS$\overset{}{\to }$*R* is denoted by $\kappa_{q}\in Z^{Q\times 1}$, with $\kappa_{q}$ = $\left[\kappa_{1},\kappa_{2},...,\kappa_{q},...,\kappa_{Q}\right]$, where $\kappa_{q}$ is a channel from IRS to *R*. The channel gain is defined as:(2)$$\kappa_{q}=cos^{2}\left(\phi \right)\frac{\sqrt{G^{R}}\alpha_{IRS}\beta_{IRS}}{d_{\kappa }}$$
where $G^{R}$ is the antenna gain at the receiver, and ϕ∈[0,π/2] is the angle of incidence. $d_{\kappa }$ is the distance from IRS$\overset{}{\to }$*R*, and $\alpha_{IRS}$ and $\beta_{IRS}$ denote the width and length of the IRS. The signal received at the receiver end is expressed as follows:(3)$$y=(\zeta_{q}\Theta \kappa_{q})x+\eta$$
where Θ∈$Z^{Q\times Q}$, and Θ is defined as:(4)$$\Theta =diag(\omega_{1}e^{j\gamma_{1}},\omega_{2}e^{j\gamma_{2}},...,\omega_{Q}e^{j\gamma_{Q}})$$
where γ = [0,2π] is the phase shift, ω = [0,1] is the amplitude of the IRS, η is the additive white Gaussian noise, and **x** is the PSK- or QAM-modulated signal.

### 3.2. Gabor Filter Network (GFN)

The Gabor filter network (GFN) is a sub-optimal feature-based pattern recognition technique that operates on a one-dimensional filter network. The GFN is very dependable in optimally detecting and differentiating the modulated signal and effectively identifying complicated signal parameters at the optimal lower and upper limits.

The Gabor filter network (GFN) parameters are calculated in [44], and the performance of these parameters is assessed in [17,34], which shows the enhanced performance for these bounds. The authors chose the lower and upper bounds in [17,34] for the GFN and evaluated the performance with these parameters. A detailed discussion on the calculation of these parameters can be found in [44], whereas the discussion on the chosen bounds’ performance is given in [17,34].

The input to the GFN is defined in Equation (Equation 3), which is the received signal. It is the first serial-to-parallel conversion to extract the distinct features from **y**. The *£* number of Gabor atom nodes is selected and can be represented as Ω = $\left[\Omega_{0},\Omega_{1},...,\Omega_{£-1}\right]$. The $\upsilon^{\mathrm{th}}$ Gabor atom node is defined as:(5)$$\Omega_{\upsilon }\left(t\right)=\frac{1}{\sqrt{\sigma_{\upsilon }}}e^{-\pi {\left(\frac{t-c_{\upsilon }}{\sigma_{\upsilon }}\right)}^{2}}\cos\left(f_{\upsilon }t\right)$$
where c, σ, and *f* are the shift, scale, and modulation parameters. The weighted output of each Gabor atom node can be expressed as:(6)$$\chi_{\upsilon }=\sum_{\upsilon =1}^{£}\left|y_{\upsilon }\underset{\Omega_{\upsilon }\left(t\right)}{\underset{︸}{\frac{1}{\sqrt{\sigma_{\upsilon }}}e^{-\pi {\left(\frac{t-c_{\upsilon }}{\sigma_{\upsilon }}\right)}^{2}}\cos\left(f_{\upsilon }t\right)}}\right|\times {℘}_{\upsilon }$$

The fitness function is the difference between the desired response ϑ(ϱ) and the output χ(ϱ) of the GFN [44]. The extracted features c, σ, and f are Gabor atom parameters, and the weights of the adaptive filter *w* are adjusted until the cost function is minimized. GF weights, which constitute the linear classification part of the GFN, are calculated by adding GF weights to the input layer, which is the feature extraction layer. According to the definition of the GF, the error is defined as follows:(7)E(ϱ)=ϑ(ϱ)−χ(ϱ)

The cost function is to minimize the mean square error, i.e.,
(8)$$J\left(\varrho \right)=min\left|\frac{1}{\Gamma }\sum_{\varrho =1}^{\Gamma }{\left[E\left(\varrho \right)\right]}^{2}\right|$$

GFNs are fed input-modulated signals during the test phase of a classifier algorithm. Once the GFN parameters are adjusted, they are fed into an adaptive filter, whose weights are optimized to reduce the error function. Now, we update the GF parameters and weights until the cost function of the adaptive filter is minimized: if the error is below the threshold, training will end; otherwise, it will continue until the cost function is minimized. The GFN parameters and adaptive filter weights are adjusted, and the error is computed. The modulation format with the smallest error is the one that will be used. Algorithm 1 lays out the procedures for constructing a Gabor filter network.
**Algorithm 1:** Proposed Gabor filter network
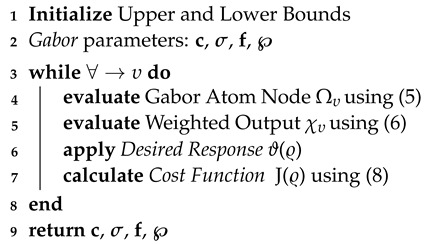


### 3.3. Optimization of Gabor Parameters and Weights

The next stage is to pick algorithms/classifiers to reliably identify the type of intercepted signal. During classifier combination, several classifiers are merged at the feature level, the data level, and the classifier or division level. A novel classification technique based on the GA has been presented for dealing with datasets with an imbalanced number of classes. The hybridized GA is intended to address the problems regarding sampling, feature subset selection, the fine tuning of traditional learning techniques, and cost matrix optimization [45]. There are several advantages to combining classifiers, and selecting a classifier with a hybrid function is the most important task, since a classifier that excels at categorizing one class may not be appropriate for another class [46].

The genetic algorithm (GA) was chosen as the optimization algorithm to minimize the cost function defined in (Equation 8) by adjusting four parameters, i.e., c, σ, f, and ***℘***. The Gabor features and weights are adjusted until the cost function ⟶ 0. Initially, the linear filter of the GFN obtains the value of its fitness function. Next, this value is input into an optimization tool utilizing one of the most well-known heuristic evolutionary computing approaches, the GA. This study developed an optimization procedure based on threshold criteria and parameter lists to identify the optimal signal utilizing global and local search techniques. Using the hybrid function effectively resolves optimization concerns. After the GA ends, the hybrid function begins operating, with the GA result as the beginning point; the hybrid function consists of the GA, GA-Fminsearch, GA-Pattern Search, GA-Fminunc, and GA-Fmincon [47,48,49]. The GA-based Gabor filter parameter and weight optimization algorithm is presented in Algorithm 2.
**Algorithm 2:** GA-based optimal Gabor parameter and weight finder.
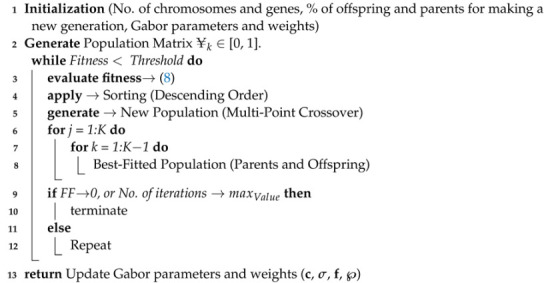



## 4. Simulation Results

The proposed classifier structure’s performance is evaluated in this section. For classification purposes, only the M-PSK and M-QAM signals are considered. The training and testing of the classifier show good classification accuracy. The proposed classifier structure is also compared with existing state-of-the-art techniques, which shows the supremacy of the classifier. The performance metric is the average classification accuracy (ACA). The simulation parameters are presented in the Table 1.

The k-fold cross-validation technique was used for the data split, which chose the optimal set of hyperparameters for the GFN. Before training the model, hyperparameters such as the number of layers, filters per layer, and the learning rate were defined. The hyperparameters that give the best results can be chosen by adjusting these hyperparameters and assessing the model’s performance on the validation set.

After optimizing the GFN, the final model was trained on the whole training set, i.e., 80%, with the specified hyperparameters. The trained model’s performance was then assessed on the test dataset, i.e., 10%, to approximate the model’s performance on unseen data. The test set must be distinct from the training and validation dataset, i.e., 10%.

### 4.1. Case 1: ACA for M-PSK Signals

The average classification accuracy of M-PSK signals, i.e., BPSK, QPSK, 16-PSK, and 64-PSK, is evaluated in this section. From Figure 2, it can be seen that the ACA approaches approximately 100% at an SNR of 5 dB. The different values of the SNR are in the range from −10→5 dB. The ACA was determined for each considered class, i.e., $\Xi =[{∁}_{1},{∁}_{2},{∁}_{3},{∁}_{4}]$, where ${∁}_{1},{∁}_{2},{∁}_{3}$, and ${∁}_{4}$ represent the BPSK-, QPSK-, 16-PSK-, and 64-PSK-modulated signals, respectively.

The ACA values for all Ξ are comprehensively presented in Table 2, which shows the classification accuracy for all classes Ξ with a fixed SNR of 10 dB. The ACA approaches approximately 100 % at an SNR of 10 dB, and as shown in Table 2, the ACA is approximately 99% for GA-based optimization. Table 2 also shows the ACA for the local search methods of GA, i.e., GA-Pattern Search (GA-PS), GA-Fmincon (GA-FM), GA-Fminsearch (GA-FS), and GA-Fminunc (GA-FU). It is evident from the results that the ACA ranges from 96% to 99% for the local search methods.

The mean square error (MSE), as stated in $R_{1,3}$ Equation (Equation 8), is presented in Figure 3. From Figure 3, the MSE with the IRS is much better than that without the IRS for all considered classes, i.e., Ξ. In addition, these modulated formats are presented with the ACA to show their performance and their discrimination with hybridized GA, GA-PS, GA-FM, GA-FS, and GA-FU. The ACA convergence tables of ${∁}_{1},{∁}_{2},{∁}_{3}$, and ${∁}_{4}$ reveal better performance than other hybridized techniques of GA for the four-class classification case of PSK signals.

### 4.2. Case II: ACA for M-QAM Signals

The classification accuracy for M-QAM signals is presented in Figure 4. For all the considered cases of QAM, the ACA is approximately 100% at an SNR of 5 dB. As seen in Figure 4, it is evident that as the SNR increases from −10 to 5 dB, the classification accuracy approaches satisfactory results. Figure 4 also shows the classification comparison of NLOS, i.e., from the GEO satellite to the user, and LOS, which is the GEO satellite to the IRS and the IRS to the user, and it is found that the ACA is much better for the LOS case as compared with NLOS. Table 3 shows the ACA comparison of local search and global optimization methods. From Table 3, it is clear that the GA-based optimization of Gabor features provides better results than local search methods.

The mean square error for the M-QAM signals is shown in Figure 5. The four-class problem for the classification of M-QAM signals is represented by $\Xi =[{∁}_{5},{∁}_{6},{∁}_{7},{∁}_{8}]$, where ${∁}_{5},{∁}_{6},{∁}_{7}$, and ${∁}_{8}$ represent the QAM-, 16-QAM-, 64-QAM-, and 256-QAM-modulated signals, respectively. As evident from Figure 5, the MSE for NLOS is more than the LOS’s.

### 4.3. Case III: Joint Classification of M-QAM and M-PSK Signals

In this simulation scenario, the eight-class problem is considered to determine the classification accuracy, where $\Xi =[{∁}_{1},{∁}_{2},{∁}_{3},...,{∁}_{8}]$, where i=1,2,3,...,8, and ${∁}_{i}$ is the same as presented in the previous sections. The confusion matrix for the classification is shown in Table 4, indicating that the LOS model’s ACA has better classification accuracy than the model without the IRS, as discussed in the previous section. Furthermore, in the confusion matrix of the eight-class problem in Table 4, the channel model is also characterized by low SNRs for the data received from the satellite with the help of the IRS. For all components involved, the QAM accuracy is 97%, while PSK has 99%, shown in the confusion matrix.

The classification accuracy for the eight-class problem is lower than in the cases discussed earlier because the proposed classifier now has eight different modulations to classify. The classification accuracy of BPSK is much better than the other modulation schemes. As seen in Table 4, the classification accuracy of higher-order QAMs is not better compared to lower-order QAM, but it is comparable.

### 4.4. Case IV: Classification of BPSK for Different Nos. of Samples

Figure 6 shows the average classification accuracy for different numbers of samples. It is evident from the figure that increasing the number of samples will result in higher classification accuracy. Figure 6 is the ACA with the IRS, and the modulation scheme considered is BPSK. The ACA is also verified for different SNRs, i.e., 0,5, and 10 dB.

In Figure 6, the ACA for the BPSK case increases, i.e., from 90% to approximately 100% at an SNR of 10 dB, with an increasing number of samples, i.e., [512,1024,2048,4096]. It is also observed that as the number of samples rises, the computational complexity of the proposed algorithm also increases, which is a trade-off between the ACA and the complexity. In this analysis, we have only chosen 1024 as the number of samples, resulting in low complexity and good classification accuracy.

### 4.5. Case V: Comparison with Existing Techniques

The proposed classifier was compared with existing state-of-the-art techniques, as shown in Table 5, and it is found that the proposed algorithm correctly classifies the GEO-satellite-enabled IRS signals. Table 5 presents the average classification accuracy, i.e., approximately 100% for the proposed classifier structure, while comparing it with the other techniques.

As stated earlier in the Contributions section, the proposed method is, to the best of our knowledge, the first of its kind; it utilizes the cascaded channel for the modulation classification, and from Table 5, it is evident that the performance of the proposed classifier is better than that of existing classifiers. The existing classifiers utilize the AWGN and fading channels, such as Rayleigh and Rician, but in our case, the considered channel is the cascaded channel.

However, the ACA comparisons presented in Table 5 were obtained directly from references and were not evaluated on the same dataset.

## 5. Discussion

Within the framework of the scheme that has been proposed, we present a proposal for a GEO satellite signal that is equipped with an IRS, together with the necessary channel modeling and system characterization. The characteristics of the GFN were evaluated to classify M-QAM and M-PSK signals. In the last step of the process, a performance evaluation of the proposed classifier structure was carried out both with and without the IRS. It used the cascaded channel for the modulation classification, and from Table 5, it is clear that the performance of the proposed classifier is better than that of existing classifiers. As mentioned earlier in the Contributions section, the proposed method is, to the best of our knowledge, the first of its kind; it employs the cascaded channel to classify the modulation. Existing classifiers make use of AWGN and fading channels such as Rayleigh and Rician, but in this scenario, the cascaded channel is the one that is being examined.

## 6. Conclusions

To transform the wireless propagation environment into a transmission medium with more desirable characteristics, an intelligent reflecting surface (IRS), as a cutting-edge technology, could potentially be implemented. In addition, IRS-enabled GEO satellite signals can be employed to extend the network capacity and improve global coverage. Based on these characteristics, this study investigated the performance of modulation classification for IRS-enabled GEO satellite signals. This GFN is optimized to improve the average classification accuracy by optimizing the Gabor parameters and weights. In general, the hybridized optimization methods perform similarly to each other. However, the GA is far superior. The primary objective of optimization was to estimate the signals of the far-field and near-field sources to determine the modulation classification. Furthermore, the simulations indicated that the classifier performs significantly better with the IRS than without the IRS. For future improvements in classification accuracy, it may be possible to refine the proposed model through genetic programming.

## Figures and Tables

**Figure 1 sensors-23-04173-f001:**
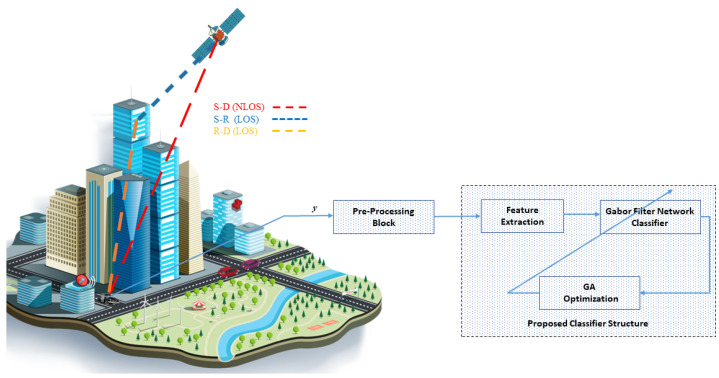
System model.

**Figure 2 sensors-23-04173-f002:**
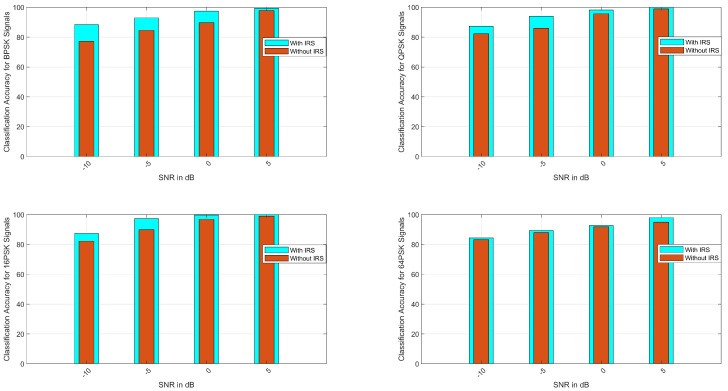
Average classification accuracy for M-PSK signals (BPSK, QPSK, 16-PSK, and 64-PSK).

**Figure 3 sensors-23-04173-f003:**
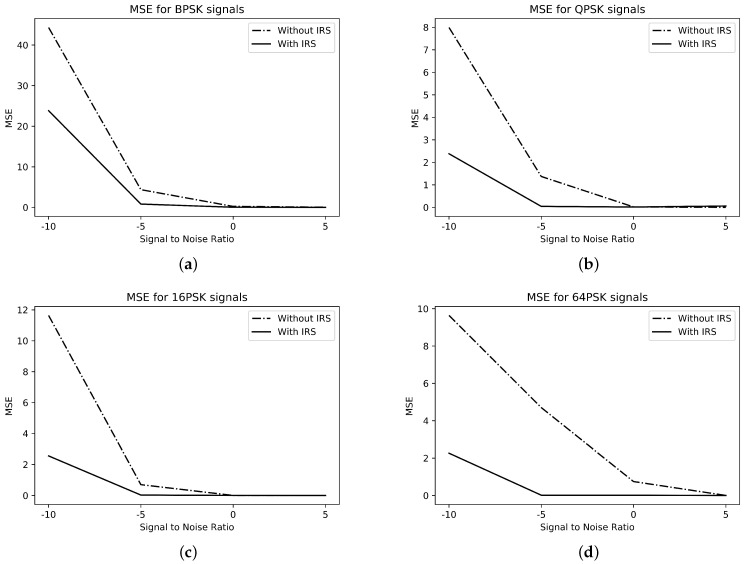
MSE for M-PSK signals. (**a**) BPSK; (**b**) QPSK; (**c**) 16-PSK; (**d**) 64-PSK.

**Figure 4 sensors-23-04173-f004:**
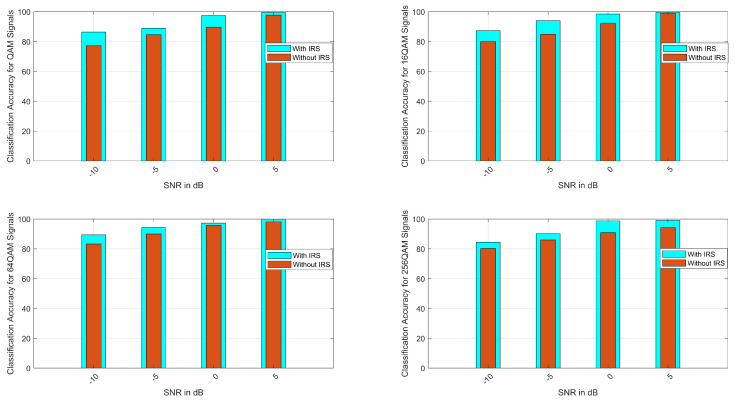
Average classification accuracy for M-QAM signals (QAM, 16-QAM, 64-QAM, and 256-QAM).

**Figure 5 sensors-23-04173-f005:**
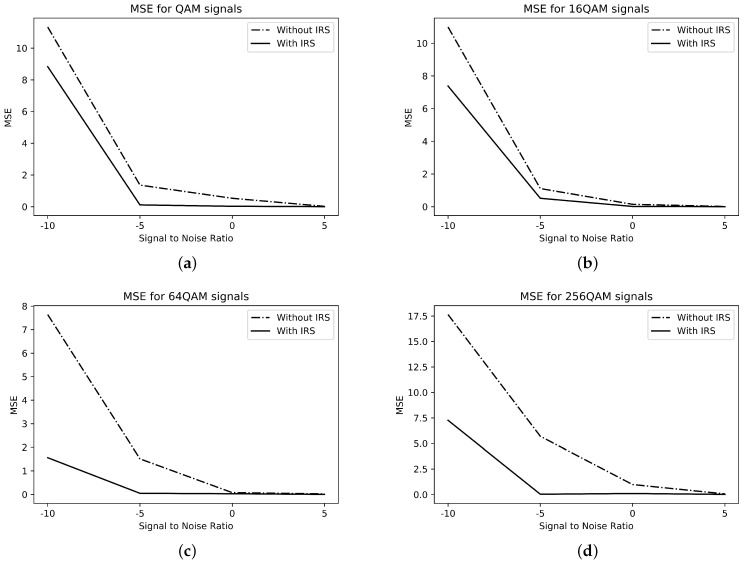
MSE for M-QAM signals. (**a**) QAM; (**b**) 16-QAM; (**c**) 64-QAM; (**d**) 256-QAM.

**Figure 6 sensors-23-04173-f006:**
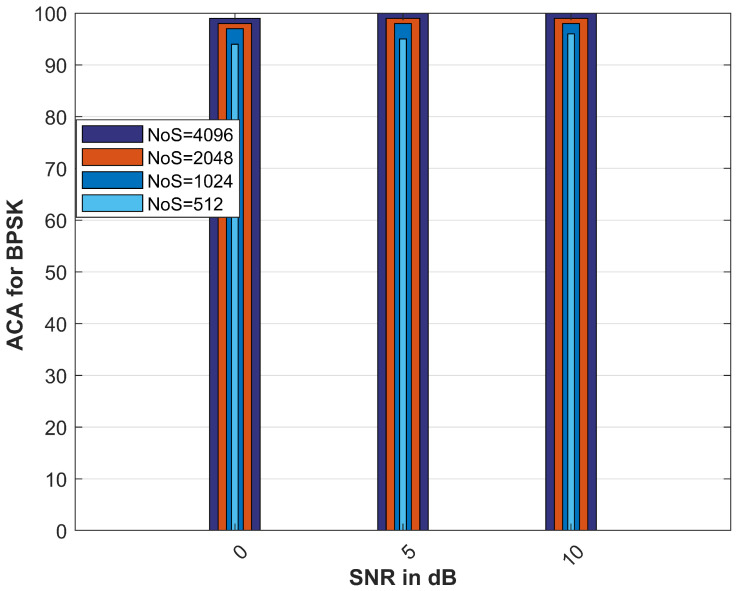
Comparison of ACA for different nos. of samples.

**Table 1 sensors-23-04173-t001:** Simulation parameters.

Parameter	Value
Solver	GA
$N_{vars}$	40
$L_{b}$	4, 1, −3.14, 0
$U_{b}$	6, 20, 3.14, 1
Fitness Limit	Infinite
Constraint tolerance	$1\times e^{-3}$
Nonlinear Constraint	Aug lag
Creation Function	GA Uniform
Population size	200×40 double
No. of Generations	100
Migration Direction	Forward
Crossover Fraction	0.8
Crossover	Multiple-Point
Functional Tolerance	$1\times e^{-6}$
Initial range	[0,1]
Scaling function	Rank
Selection	Best-Fitted
Elite count	0.05×Population Size
Mutation function	Adaptive Feasible
Problem Type	Bound Constraints

**Table 2 sensors-23-04173-t002:** ACA for M-PSK signals with IRS.

SNR = 5 dB	GA	GA-PS	GA-FM	GA-FS	GA-FU
**BPSK**	99.37	97.20	96.30	96.43	97.76
**QPSK**	99.98	98.36	98.70	97.93	98.36
**16-PSK**	99.68	99.12	98.34	98.60	99.21
**64-PSK**	97.98	96.31	95.44	96.72	97.11

**Table 3 sensors-23-04173-t003:** ACA for M-QAM signals with IRS.

SNR = 5 dB	GA	GA-PS	GA-FM	GA-FS	GA-FU
**QAM**	99.48	94.68	98.93	97.23	96.75
**16-QAM**	99.52	91.23	98.31	98.11	98.00
**64-QAM**	99.95	97.23	98.45	97.54	98.75
**256-QAM**	99.02	91.45	97.23	96.55	97.34

**Table 4 sensors-23-04173-t004:** Confusion matrix for 8-class problem.

SNR = 0 dB	QAM	16-QAM	64-QAM	256-QAM	BPSK	QPSK	16-PSK	64-PSK
**QAM**	**97.23**	1.77				1.0		
**16-QAM**	1.50	**96.33**	1.67				0.5	
**64-QAM**	1.42	2.18	**95.48**					0.92
**256-QAM**	1.14	2.20	2.54	**94.12**				
**BPSK**					**98.87**	1.13		
**QPSK**		1.55				**98.45**		
**16-PSK**					1.21	1.29	**96.78**	0.72
**64-PSK**						2.2	0.57	**97.23**

**Table 5 sensors-23-04173-t005:** Comparison with existing techniques.

Reference	Method and Algorithm	Channel Type	Modulation Type	ACA
**[50]**	Likelihood Function (kurtosis)	AWGN	ASK, FSK, PSK	98.8%
**[51]**	CNN	AWGN	LFM, BPSK	98%
**[52]**	NCMBFW	AWGN	ASK, FSK, PSK	99.1%
**[53]**	Fusion of Features	AWGN	PSK, FSK, QAM	99%
**[54]**	Wavelet transform	Fading + Gaussian noise	QAM, PSK, FSK	97%
**[55]**	GA-SVM	Rician, Rayleigh, AWGN	M-QAM	96%
**[17]**	Neural-network-based SVM	AWGN, Rayleigh	QAM	98 %
**[20]**	GFN + ABC	Rician, Rayleigh, AWGN	QAM, PSK, FSK	97%
**[56]**	Feature Fusion	Multi-Channel	QAM, PSK	95%
**[57]**	H-SOM	-	H-SOM, QAM, PSK	97.9%
**2023**	Proposed	Cascaded channel	QAM, PSK	99.7 %

## Data Availability

Not applicable.

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
