# Peer review of "Optimized Classification of Intelligent Reflecting Surface (IRS)-Enabled GEO Satellite Signals"

_sensors, 2023, doi:10.3390/s23084173_

Round 1
Reviewer 1 Report
This paper introduces a novel classification schema for intelligent reflecting surface (IRS)-enabled GEO satellite signals, utilizing a Gabor filter network that has been optimized through a proposed genetic algorithm. The effectiveness of this approach is evaluated through simulations, which include a range of M-PSK and M-QAM signals with varying SNRs and sample sizes, as well as selected benchmarks.
Automatic modulation classification is a crucial challenge in practical applications, and the literature review and problem description in this paper are easy to comprehend for readers interested in the topic. The system model for both line-of-sight (LOS) and non-line-of-sight (NLOS) scenarios is described in detail, providing a comprehensive overview of the proposed approach. The presentation of the Gabor filter network (GFN) and optimization algorithm is clear. While the simulation results for different cases are presented, there is room for improvement in terms of further analysis and evaluation.
Major comments:
1. Thresholds are referenced multiple times in the paper when introducing the Gabor filter network (GFN) and the optimization algorithm. To provide readers with a clear understanding of this aspect of the proposed approach, it would be beneficial to include a more detailed explanation of how the thresholds are chosen. This would enhance the overall clarity and transparency of the methodology, ensuring that readers can fully comprehend the GFN and optimization process.
2. In subsections 4.1 and 4.2, the simulation results are presented for a limited range of SNRs. To provide a more comprehensive analysis, it would be beneficial to extend the simulations to a broader range of SNRs, such as -20 dB to 20 dB. This would allow for a more thorough evaluation of the proposed approach and provide readers with a more complete understanding of its performance under varying signal-to-noise ratios.
3. The gain of classification accuracy with IRS is not consistent with varying SNRs in Figure 3 and 4. IRS helps in low SNR sometimes while sometimes in high SNR. A detailed discussion would be beneficial to explain why the effectiveness of the IRS may vary with the SNR. In addition, it would be beneficial to present a confidence interval for the achieved classification accuracy (ACA) to provide a measure of the statistical uncertainty associated with the results. This would help readers to interpret the accuracy figures and evaluate the reliability of the proposed approach.
4. Table 4 appears to have many missing entries, which could potentially limit its usefulness for readers. It is important to ensure that all relevant information is included in the table and that it accurately represents the simulation results.
5. It is crucial to provide a detailed explanation of how the data is split for optimizing the Gabor filter network (GFN) and for testing the proposed approach to ensure the reliability and comparability of the results. Specifically, it is important to clarify how the training, validation, and testing data is split and to ensure that the same data split is used when comparing the proposed approach with other methods in subsection 4.5. This will help to avoid any biases or inconsistencies that could impact the accuracy and generalizability of the results.
6. The observed linear decrease in mean squared error (MSE) as the signal-to-noise ratio (SNR) increases in Figures 3 and 5 is not entirely clear and warrants further explanation.
Author Response
Note: In the revised manuscript, [R1,2] corresponds to changes made in response to comment 2 of Reviewer 1
Reviewer 1
We thank the reviewer for the time, effort, and valuable comments. We believe that all comments are constructive and contribute to enhancing the readability of our paper and enriching its contents. Therefore, all comments have been addressed appropriately. Please find the response and amendments made to your comments below.
This paper introduces a novel classification schema for intelligent reflecting surface (IRS)-enabled GEO satellite signals, utilizing a Gabor filter network that has been optimized through a proposed genetic algorithm. The effectiveness of this approach is evaluated through simulations, which include a range of M-PSK and M-QAM signals with varying SNRs and sample sizes, as well as selected benchmarks.
Automatic modulation classification is a crucial challenge in practical applications, and the literature review and problem description in this paper are easy to comprehend for readers interested in the topic. The system model for both line-of-sight (LOS) and non-line-of-sight (NLOS) scenarios is described in detail, providing a comprehensive overview of the proposed approach. The presentation of the Gabor filter network (GFN) and the optimization algorithm is clear. While the simulation results for different cases are presented, there is room for further analysis and evaluation.
1) Thresholds are referenced multiple times in the paper when introducing the Gabor filter network (GFN) and the optimization algorithm. To provide readers with a clear understanding of this aspect of the proposed approach, it would be beneficial to include a more detailed explanation of how the thresholds are chosen. This would enhance the overall clarity and transparency of the methodology, ensuring that readers can fully comprehend the GFN and optimization process.
The Gabor filter network (GFN) parameters are calculated in [1], and the performance of these parameters is assessed in [2], [3], which shows the enhanced performance for these bounds. Authors chose these lower and upper bounds of [2], [3] for GFN and evaluated the performance on these parameters. A detailed discussion on the calculation of these parameters can be found in [1], whereas the discussion on the performance for the chosen bounds are given in [2], [3].
2) In subsections 4.1 and 4.2, the simulation results are presented for a limited range of SNRs. To provide a more comprehensive analysis, it would be beneficial to extend the simulations to a broader range of SNRs, such as -20 dB to 20 dB. This would allow for a more thorough evaluation of the proposed approach and provide readers with a more complete understanding of its performance under varying signal-to-noise ratios.
The revised manuscript presents the average classification accuracy (ACA) for the M-PSK, and M-QAM signals with IRS in Tables 2 and 3, respectively. The ACA is approximately approached 100% at 5dB of SNR for both modulation scenarios. So there is no need to increase the SNR by more than 5dB. Moreover, the article’s main contribution is to show the accuracy at lower SNRs.
However, in a satellite communication system, usually, the SNR is high, i.e., more than 5dB. As we have considered the Geo-Satellite signals, there is no need to evaluate the classifier performance on very low SNRs, i.e., -20dB. Moreover, per the reviewer’s comment, the classification is below 60% at SNRs less than -10 dB.
3) The gain of classification accuracy with IRS is not consistent with varying SNRs in Figure 3 and 4. IRS helps in low SNR sometimes while sometimes in high SNR. A detailed discussion would be beneficial to explain why the effectiveness of the IRS may vary with the SNR. In addition, it would be beneficial to present a confidence interval for the achieved classification accuracy (ACA) to provide a measure of the statistical uncertainty associated with the results. This would help readers to interpret the accuracy figures and evaluate the reliability of the proposed approach.
The authors agree with the reviewer, and it is true that IRS helps in low SNR sometimes while sometimes in high SNR. As seen from Figure 2 and Figure 4 from the manuscript, All the considered modulation formats constellations are independent; for example, for QAM, there are four constellation points, and as we increase the SNR from -10 dB to 5 dB, the signal dominants the noise, there is a rapid increase in the classification accuracy but at higher SNRs, maximum classification accuracy is achieved so there is a slight improvement in the classification
4) Table 4 appears to have many missing entries, which could potentially limit its usefulness for readers. It is important to ensure that all relevant information is included in the table and that it accurately represents the simulation results.
As per the reviewer’s comment, Table 4 of the manuscript has been revised and can be seen in response to the reviewer’s comments as Table 1. In general practice, there is only a display of the accuracy, but as per the reviewer’s comment, the misclassification is also included in the Table.
From the Table, it is clear that the classification accuracy of the proposed 8-class problem is much better at lower SNRs.
5) It is crucial to provide a detailed explanation of how the data is split for optimizing the Gabor filter network (GFN) and testing the proposed approach to ensure the reliability and comparability of the results. Specifically, it is important to clarify how the training, validation, and testing data is split and to ensure that the same data split is used when comparing the proposed approach with other methods in subsection 5. This will help to avoid any biases or inconsistencies that
TABLE I: R1,4 Confusion Matrix for 8-Class Problem
|
SNR=0dB |
QAM |
16QAM |
64QAM |
256QAM |
BPSK |
QPSK |
16PSK |
64PSK |
|
QAM |
97.23 |
1.7 |
|
|
|
1.0 |
|
|
|
16QAM |
1.50 |
96.33 |
1.67 |
|
|
|
0.5 |
|
|
64QAM |
1.42 |
2.18 |
95.48 |
|
|
|
|
0.92 |
|
256QAM |
1.14 |
2.20 |
2.54 |
94.12 |
|
|
|
|
|
BPSK |
|
|
|
|
98.87 |
1.13 |
|
|
|
QPSK |
|
1.55 |
|
|
|
98.45 |
|
|
|
16PSK |
|
|
|
|
1.21 |
1.29 |
96.78 |
0.72 |
|
64PSK |
|
|
|
|
|
2.2 |
0.57 |
97.23 |
could impact the accuracy and generalizability of the results.
The k-fold cross-validation technique is used for data split, which chooses the optimal set of hyperparameters for the GFN. Before training the model, hyperparameters such as the number of layers, filters per layer, and the learning rate are defined. The hyperparameters that give the best results may be chosen by adjusting these hyperparameters and assessing the model’s performance on the validation set.
After optimizing the GFN, the final model is trained on the whole training set, i.e., 80% with the specified hyperparameters. The trained model’s performance is then assessed on the test data set, i.e., 10%, to approximate the model’s performance on unseen data. The test set must be distinct from the training and validation dataset, i.e., 10%. In Table 5, the authors of [4] have used the same training, validation, and testing data, and the authors of [5] used 80% for training and 20% for testing, whereas other authors did not disclose the splitting of the data.
6) The observed linear decrease in mean squared error (MSE) as the signal-to-noise ratio (SNR) increases in Figures 3 and 5 is not entirely clear and warrants further explanation.
The results in Figure 3 and Figure 5 are not linear, as the results are generated based on the SNR of -10 dB, -5 dB, 0 dB, 5 dB, and 10 dB. The results are generated based on these SNRs; the x-axis range may confuse the readers and be interpreted as linear results. The authors will revise the x-axis range to mitigate ambiguity.
TABLE II: Comparison with Existing Techniques
|
References |
Method & Algorithm |
Channel Type |
Modulation Type |
ACA |
|
[6] |
Likelihood Function (kurtosis) |
AWGN |
ASK, FSK, PSK |
98.8% |
|
[5] |
CNN |
AWGN |
LFM, BPSK |
98% |
|
[7] |
NCMBFW |
AWGN |
ASK, FSK, PSK |
99.1% |
|
[8] |
Fusion of Features |
AWGN |
PSK,FSK, QAM |
99% |
|
[9] |
Wavelet transform |
Fading + Gaussian Noise |
QAM,PSK,FSK |
97% |
|
[10] |
GA-SVM |
Rician, Rayleigh, AWGN |
M-QAM |
96% |
|
[11] |
Neural Network based SVM |
AWGN, Rayleigh |
QAM |
98 % |
|
[4] |
GFN+ABC |
Rician, Rayleigh, AWGN |
QAM, PSK, FSK |
97% |
|
R2,4 [12] |
Deep Learning Features Fusion |
Multi Channel |
QAM, PSK |
95% |
|
R2,4 [13] |
H-SOM |
- |
H-SOMQAM, PSK |
97.9% |
|
2023 |
Proposed |
Cascaded Channel |
QAM, PSK |
99.7% |

Reviewer 2 Report
1. The authors need to provide more theoretical background.
2. The simulation studies must be validated through another simulation tool and the authors are advised to compare the results.
3. Objective and Novelty of the work can be elaborated and also the design methodology section must be explained in more detail.
4. A few more recent last 2 years references must be included and those articles should be compared.
Author Response
Note: In the revised manuscript, [R2,2] corresponds to changes made in response to comment 2 of Reviewer 2.
Reviewer 2
We thank the reviewer for the time, effort, and valuable comments. We believe that all comments are constructive and contribute to enhancing the readability of our paper and enriching its contents. Therefore, all comments have been addressed appropriately. Please find the response and amendments made to your comments below.
1) The authors need to provide more theoretical background.
The theoretical background related to the article has been added to the revised manuscript as suggested by the reviewer.
2) The simulation studies must be validated through another simulation tool, and the authors are advised to compare the results.
The simulations for the research are performed on Matlab (simulation tool), where the weight of the Gabor filter (GF) is optimized using the genetic algorithm (GA) and various variants of GA. The weights are optimized so that the classification error is minimized.
The authors appreciate the reviewer’s comments for evaluating the simulations on different tools. However, the authors believe that the simulation tool does not impact the performance because the algorithm used to optimize the GF’s weights is GA and its variants. There is no difference in the algorithm between different tools, but this will require additional time.
Accordingly, the authors plan to use a variety of tools in future research in light of the reviewer’s comments.
3) Objective and Novelty of the work can be elaborated, and the design methodology section must be explained in more detail.
The objectives and novelty of the work have been updated in article section 1.1, contribution section. The design methodology has been comprehensively explained in section 3, i.e., proposed classifier
4) A few more recent last two years’ references must be included, and those articles should be
As per the reviewer’s comment, the proposed model has been compared with the two latest references and Incorporated in the revised manuscript.
REFERENCES
[1]S. A. Ghauri, I. M. Qureshi, T. A. Cheema, and A. N. Malik, “A novel modulation classification approach using gabor filter network,” The Scientific World Journal, vol. 2014, 2014.
[2]S. I. H. Shah, A. Coronato, S. A. Ghauri, S. Alam, and M. Sarfraz, “Csa-assisted gabor features for automatic modulation classification,” Circuits, Systems, and Signal Processing, pp. 1–23, 2022.
[3]S. I. H. Shah, S. Alam, S. A. Ghauri, A. Hussain, and F. A. Ansari, “A novel hybrid cuckoo search-extreme learning machine approach for modulation classification,” IEEE Access, vol. 7, pp. 90 525–90 537, 2019.
[4]S. AlJubayrin, M. Sarfraz, S. A. Ghauri, M. R. Amirzada, and T. M. Kebedew, “Artificial bee colony based gabor parameters optimizer (abc-gpo) for modulation classification,” Computational Intelligence and Neuroscience, vol. 2022, 2022.
[5]M. Zhang, M. Diao, and L. Guo, “Convolutional neural networks for automatic cognitive radio waveform recognition,” IEEE Access, vol. 5, pp.
11 074–11 082, 2017.
[6]P. K. HL and L. Shrinivasan, “Automatic digital modulation recognition using minimum feature extraction,” in 2015 2nd International Conference on Computing for Sustainable Global Development (INDIACom). IEEE, 2015, pp. 772–775.
[7]N. Daldal, K. Polat, and Y. Guo, “Classification of multi-carrier digital modulation signals using ncm clustering based feature-weighting method,”
Computers in Industry, vol. 109, pp. 45–58, 2019.
[8]S. Zheng, P. Qi, S. Chen, and X. Yang, “Fusion methods for cnn-based automatic modulation classification,” IEEE Access, vol. 7, pp. 66 496–66 504, 2019.
[9]W. Li, Z. Dou, L. Qi, and C. Shi, “Wavelet transform based modulation classification for 5g and uav communication in multipath fading channel,”
Physical Communication, vol. 34, pp. 272–282, 2019.
[10]N. Bany Muhammad, S. Ghauri, M. Sarfraz, and S. Munir, “Genetic algorithm assisted support vector machine for m-qam classification,” 2020.
[11]S. I. H. Shah, A. Coronato, S. A. Ghauri, S. Alam, and M. Sarfraz, “Csa-assisted gabor features for automatic modulation classification,” Circuits, Systems, and Signal Processing, vol. 41, no. 3, pp. 1660–1682, 2022.
[12]H. Han, Z. Yi, Z. Zhu, L. Li, S. Gong, B. Li, and M. Wang, “Automatic modulation recognition based on deep-learning features fusion of signal and constellation diagram,” Electronics, vol. 12, no. 3, p. 552, 2023.
[13]Z. Li, Q. Wang, Y. Zhu, and Z. Xing, “Automatic modulation classification for mask, mpsk, and mqam signals based on hierarchical self-organizing map,” Sensors, vol. 22, no. 17, p. 6449, 2022.
Round 2
Reviewer 1 Report
In general, most of the comments have been taken into account. However, there are several issues that should be addressed.
Major comments:
1. As mentioned in the authors’ response, R_{1,5}, the ACA comparison with existing techniques is actually not an apple-to-apple comparison because the chosen techniques are not evaluated on the same testing dataset. The best recommended course of action is to re-run different techniques to train and evaluate on the same training and testing dataset. If the ACA numbers are directly acquired from the references in Table 5, the minimal recommended action is to say explicitly that the achieved ACA of different techniques are acquired directly from the references and are not evaluated on the same testing dataset.
Minor comments:
1. The sum of the first row of the updated Table 4 is not equal to 1.
2. In row 272, the mean square error is referred to as Eq. (6). Should it be Eq. (8)?
Author Response
We thank the reviewer for the time, effort, and valuable comments. We believe that all comments are constructive and contribute to enhancing the readability of our paper and enriching its contents. Therefore, all comments have been addressed appropriately. Please find the response and amendments made to your comments below.
In general, most of the comments have been taken into account. However, there are several issues that should be addressed.
1) As mentioned in the authors’ response, R1,5, the ACA comparison with existing techniques is actually not an apple-to-apple comparison because the chosen techniques are not evaluated on the same testing dataset. The best
recommended course of action is to re-run different techniques to train and evaluate on the same training and testing dataset. If the ACA numbers are directly acquired from the references in Table 5, the minimal recommended action is to say explicitly that the achieved ACA of different techniques are acquired directly from the references and are not evaluated on the same testing dataset.
The authors highly appreciate the reviewer’s highlighting of an important point. The authors agree with the reviewer that this is not an apple-to-apple comparison. The proposed classification of IRS-enabled GEO satellite signals is a
novel work not addressed in the literature to the best of the author’s knowledge.
However, the ACA comparisons presented in Table 5 were obtained directly from references and were not evaluated on the same dataset. This is also added in the text.
2) The sum of the first row of the updated Table 4 is not equal to 1.
There is a typo mistake, corrected in the revised version of manuscript.
3) In row 272, the mean square error is referred to as Eq. (6). Should it be Eq. (8)?
Yes agree with the reviewer comment and corrected in the revised version.